# Simultaneous Measurement of Strain and Displacement for Railway Tunnel Lining Safety Monitoring

**DOI:** 10.3390/s24196201

**Published:** 2024-09-25

**Authors:** Jun Li, Yuhang Liu, Jiarui Zhang

**Affiliations:** College of Safety Science and Engineering, Xi’an University of Science and Technology, Xi’an 710054, China; edreiliu@outlook.com (Y.L.); jiaruiziy@163.com (J.Z.)

**Keywords:** deformation, submillimeter, structural health monitoring, railway tunnel

## Abstract

This paper proposes a dual-parameter strain/displacement safety monitoring technology for railway tunnel lining structures. An integrated monitoring system with FBG (Fiber Bragg grating) and VDM (video displacement meter) components was used to monitor both the strain and deformation of the tunnel cross-section. Initially, a comprehensive experimental study was carried out using FBG strain sensors with temperature-compensated grating. The temperature-compensated grating was used to further improve the monitoring accuracy. The data show that the stability and accuracy were better than the traditional electronic strain sensor. Secondly, high-precision and multipoint monitoring of railway tunnel lining deformation was achieved by using VDM technology. Three months of case study results taken from the Gansu Railway Tunnel in China demonstrated a tunnel cross-section strain accuracy for microstrain and crown deformation at the submillimeter level, respectively. The technology provides a new high-precision way to monitor the condition of tunnel lining structures.

## 1. Introduction

Railway tunnels are important underground transport facilities that provide comfort to people. However, various structural problems such as cracking, settlement, seepage, and voids are inevitable during the operation of railway tunnels, due to factors such as their design, the materials used, and the geological environment. The severity of structural damage can significantly affect the safety and stability of railway tunnel structures [1,2,3]. As a result, the structural health monitoring and diagnosis of railway tunnels have become increasingly important worldwide. Researchers have proposed effective solutions for the normal operation of railway tunnels through processes such as immediate monitoring, data analysis, damage identification, and safety condition assessment of railway tunnel structures.

At present, tunnel structure health monitoring methods in railway operations can be mainly divided into those relying on contact and noncontact measurements. Contact measurements can obtain fundamental data on the surface or inside the tunnel via manual deployment or installation of sensors on the monitoring surface, such as electronic sensors [4], Fiber Bragg grating (FBG) sensors [5], etc. Conversely, noncontact measurements are carried out by assessing the potential monitoring area via optical or structural characteristics to acquire monitoring data indirectly with high precision, using tools such as total stations [6], laser displacement meters [7], terrestrial laser scanning (TLS) [8], etc. Although manual inspection methods can ensure that the cracks and displacements in tunnels are within a controllable range, their drawbacks are obvious: low working efficiency, large recording deviation, long time data delay, and incapability to achieve 24/7 monitoring. Electronic vibrating strain sensors [9,10] can achieve automatic monitoring, but they are unstable. Fiber optic sensing technology [11,12] is passive in nature, immune to electromagnetic interference, and possesses high measurement accuracy and ease of networking. Total stations [13,14] utilize the light reflected from the prisms located in the far end to obtain tunnel deformation information with accuracy within millimeters during the construction period; however, the monitoring distance is limited and automatic monitoring is difficult to achieve. Intelligent total stations [15] solve the problem of manual calibration, but remain unsuitable for all-weather monitoring. Although tunnel detection systems based on intelligent robots [16,17] have realized automatic detection capability, the cost is still high, and long-term uninterrupted monitoring is not achievable. TLS technology [18,19] can achieve a displacement accuracy of 1 mm/10 m; however, the technology is expensive and not capable of continuous monitoring. OFDR distributed fiber sensing techniques [20] have high spatial resolution and measurement accuracy, but their stability is low under high-temperature and high-pressure conditions. Photogrammetry [21,22] is able to remotely assess and monitor targets without touching the tunnel structure, and is not affected by the vibration of passing trains in the tunnel, which reduces the risk of accidents. In addition, photogrammetry is generally more cost effective than the traditional measurement methods.

In this paper, an operational railway tunnel lining structure health monitoring system based on the combination of FBG and photogrammetry technology is proposed. By installing FBG sensors and VDM at key positions on the lining surface of a railway tunnel cross-section in Gansu, China, multipoint high-precision strain and displacement are quantitatively analyzed. Finally, the system performance is verified by short and long-term monitoring data.

## 2. Measurement Principle

### 2.1. Fiber Bragg Grating Monitoring Technology

The principle of fiber grating monitoring is based on Bragg grating reflection, where light in a certain frequency range is reflected by a periodic refractive index modulation along an optical fiber. When FBGs are subjected to external perturbations, such as temperature or strain changes, their Bragg wavelength shifts, which can be detected and measured [23,24,25].

In practical applications, this technology is used to monitor tunnel structural strain and prevent fire. Measurements are usually made by embedding or cascading one or more FBGs onto the structure being monitored [26,27,28].

Normally, all the incident light will pass through FBG without being affected; only the light of a specific wavelength will return to the original direction after being reflected at the FBG, as can be seen from Figure 1. According to the mode coupling theory, the wavelength that satisfies the following relationship can be reflected back:(1)$$\lambda_{B}=2n_{eff}\Lambda$$
where $\lambda_{B}$ is the central wavelength of the fiber grating; Λ is the period of the grating; and $n_{eff}$ is the effective refractive index of the fiber core.

It can be seen from Equation (1) that the reflected central wavelength $\lambda_{B}$ is related to the grating period Λ and the core refractive index $n_{eff}$. The measurement of the required parameters can be carried out by detecting the change in the center wavelength of FBG.

### 2.2. Photogrammetry Technique

This paper uses a CCD camera to collect and recognize image information of the tunnel cross-section. These images are then post-processed to obtain key parameters so that the position and size information of the monitored object can be calculated with high precision [22]. As shown in Figure 2, the general idea is to use the template matching algorithm to calculate and process the structure vibration video collected by the camera and convert the displacement into a physical coordinate system. The realization of this function is mainly based on two advanced image processing technologies: the normalized cross-correlation algorithm [29] and the subpixel technology [30].

First, the captured video is read as a series of greyscale images. Then, a selected region on the first or specified frame image is used as a template. A template matching technique is applied to locate the selected template image in a continuous image and obtain a pixel coordinate system, calculate the displacement within the pixel coordinates, and obtain the displacement of the surveillance structure by a conversion factor. To reduce the computation time, the search area can be limited to a predefined region of interest (ROI) close to the template in the image. The execution of the algorithm is shown in Figure 2.

Normalization of the image is achieved by subtracting the mean of all pixels from each pixel and further dividing by the standard deviation of all images so that all the pixel values can be placed into the same scale for subsequent processing. Because of the fluctuation of pixel intensity and contrast in the relevant area caused by the light intensity and the reflectivity of the object, data collection becomes difficult. Normalization is immune to fluctuation, compensates the relative deviation, and obtains stable image features. The normalized product correlation algorithm uses the quasi-side of the real-time image and the reference image to choose the maximum correlation coefficient (i.e., the highest similarity) as the best matching location. As shown in Equation (2), its correlation matrix can be expressed as follows:(2)$$\theta =\left[\begin{array}{cccc} \rho \left(0,0\right) & \rho \left(0,1\right) & \cdots & \rho \left(0,W-M\right)\\ \rho \left(1,0\right) & \cdots & \cdots & \cdots \\ \vdots & \vdots & \vdots & \vdots \\ \rho \left(H-N,0\right) & \cdots & \cdots & \rho \left(H-N,W-M\right) \end{array}\right]$$

Assuming that the reference image is *I*, with its size being *W***H*, the real-time image is *R*, with the size of *M***N*, then the corresponding correlation coefficient is (*W* − *M* + 1)*(*H* − *N* + 1). The normalized product line pipe coefficients for real-time and reference images at point (*x*, *y*) in the coordinate system are defined as follows:(3)$$C_{NCC}\left(x,y\right)=\frac{\sum_{i=0}^{M-1}\sum_{j=0}^{N-1}I\left(x+i,y+j\right)T\left(i,j\right)}{\sqrt{\sum_{i=0}^{M-1}\sum_{j=0}^{N-1}I{\left(x+i,y+j\right)}^{2}}\sqrt{\sum_{i=0}^{M-1}\sum_{j=0}^{N-1}T{\left(i,j\right)}^{2}}}$$

## 3. Optical Video Displacement Meter Test Experiment

To ensure that the measurement accuracy of the camera meets the requirements of the actual tunnel lining structure monitoring applications, comparison experiments, illuminance testing experiments, and fog testing experiments are carried out using a commercial camera (adoption of grey dot camera with pixel unit up to 4.8 µm and maximum resolution/frequency of 1280*1024 @ 150FPS (8-bit)), aiming to eliminate the influence of environmental effect on the measurement results, to determine the clarity of the field of vision and light intensity that should be maintained during the measurement, and to ensure a stable and reliable measurement data.

### 3.1. Instrument Accuracy Comparison

The camera and laser displacement meter (LD-40) are fixed on the same bracket, the target bracket of both is fixed on the same lifting platform, while the height gauge is fixed on one side of the lifting platform. The height change value of the lifting platform can be read through the change of the height gauge reading. The laser displacement meter was placed in a position about 10 m away from the target. The angle of view was adjusted to ensure a complete match between the target and the displacement meter, as shown in Figure 3. The movement of the lifting platform of the target was increased by 2 mm and steadily reached 20 mm, and then it stabilized the target by 2 mm each time and dropped to 0 mm. After each adjustment, the data were stabilized for 5 min, and then displacement data were recorded three consecutive times and the averaged value was taken.

As shown in Figure 4, the experimental data indicate that both the laser displacements and video displacements provide highly accurate measurements. The laser displacement meter boasts a minimum recognition accuracy of 0.01 mm, while the VDM offers a higher recognition range and a minimum accuracy of 0.001 mm. At the same time, the data indicate a discrepancy between the measured displacement value by the displacement meter and the actual displacement, wherein the maximum error of the VDM during testing is 0.082 mm, primarily remaining within 0.100 mm. Conversely, the laser displacement meter has a maximum error of 0.49 mm and can be only maintained within 0.50 mm, resulting in the VDM having superior measurement accuracy compared to the laser displacement meter. At the same time, the laser displacement meter used in the experiment shows an increase in error with the target position. Based on its construction and monitoring principle, displacement changes can occur due to changes in the original straight-line displacement. These changes evolve into a trigonometric function, leading to displacement conversion errors, which is also a major drawback of the laser displacement meter.

### 3.2. Illumination Experiment

The experiment setup is shown in Figure 5. A place without human interference and external damage was selected and fixed on the ground with epoxy through the base of the VDM to ensure a relatively stabilized environment. At the same time, two prepared two-dimensional codes of the target are fixed on the upgrade platform attached to the side micrometer, and then the upgrade platform is fixed on the ground with epoxy. We adjusted the matching angle between the VDM and the target to make it work normally. From the morning, the illuminance instrument was used to collect the illuminance value every 1 h. During the collection process, the displacement of the VDM was recorded. The total test lasted for 14 h, from 7 a.m. to 9 p.m., in April, during which the camera and the lifting platform target were kept unchanged and the final test data were summarized.

By default, when the target of the camera and lifting platform do not move, the output from VDM should remain near zero. As can be seen from Figure 6, for illuminance of VDM and displacement data chart, during 13 h from 8:00 to 21:00 on the same day, the ambient illuminance changes obviously, with the highest value reaching nearly 7000 Lx around 14:00 and the lowest illuminance only in single digits. There is a certain correlation characteristic between the tested displacement and illuminance. It is clear that the displacement value almost does not change during 8:00~11:00 and 15:00~18:00, while the illuminance data and displacement variation change significantly during 11:00~15:00. It can be seen that the displacement fluctuation is larger than the change of light illuminance (about 0.08 mm). The reason may be due to the direct sunlight of the target when the illuminance is at its maximum during the testing period, resulting in large displacement fluctuation. When the light flux is lower than the single digit, the displacement fluctuation is greater. When the light intensity is lower than the single digit, the displacement fluctuation difference is up to 0.16 mm, and the displacement accuracy is far lower than that in the conventional environment. The main reason for the accuracy variation is that the illumination intensity is too weak. The acquired target image will become blurred, and the accuracy of the shot image is different from that with sufficient light. Therefore, in practical application, either direct sunlight exposure or extremely weak illumination working environment should be avoided. The current status quo of the VDM in the tunnel does not rely on direct sunlight as a source of illumination. The actual railway tunnels have an illumination of over a hundred digits and are more stable. Consequently, it is planned to use a cold light source in future applications of the VDM to supplement the luminous flux and improve the accuracy of displacement test data.

### 3.3. Fog Experiment

The experiment setup is shown in Figure 7. Under the indoor constant temperature, the VDM base is hardwired and fixed onto the optical platform with screws to ensure the stability of the camera. At the same time, two target two-dimensional codes are fixed on the upgrade platform attached to the side of a micrometer, and then the upgrade platform is fixed onto another optical platform. We adjusted the matching angle between the VDM and the target to make it work normally. We installed an atomizer at the bottom before target 2, so that it generates fog interference at the position of target 2, which makes target 2 seriously shielded by fog. The lifting platform of the target rises steadily to 20.00 mm and then decreases steadily to 0 mm with a step of 2.00 mm each time. After each adjustment, the platforms remained stable for 5 min and then recorded displacement data, and the average value was taken to obtain the experimental data.

Figure 8 illustrates the differences in displacement data error between target 1 and target 2 in a fog interference environment. The test displacement of target 1 demonstrates stronger consistency with the displacement of the upgrading stage, with a maximum error of only 0.06 mm. However, target 2 displays larger discrepancies between measured displacement values and the actual displacement, with a maximum error of around 0.16 mm. In addition, it is worth noting that under the presence of fog, there were multiple mutations observed in target 2. Its stability and accuracy are considerably lower compared to target 1 or in a standard environment. These findings suggest that the accuracy of the VDM is significantly affected by the presence of fog. For target 2, the measurement of displacement error is greater than expected, mainly due to the seriousness of the fog on its masking, which obscures the structure of the surface features. Additionally, the fog itself is also in a state of flow, which further contributes to multiple factors leading to a larger error in measurement. For the experiments on fog testing, the accuracy of the VDM is reduced due to the challenging nature of the fog environment. Although the program algorithms have undergone some degree of noise reduction processing of the data weakened by the interference of the fog, its impact is still prominent. Therefore, it is advisable to avoid using this system in foggy or humid environments such as lakes and valleys. Due to the higher humidity in Northwest China, the overall environment is dry. As the tunnel has been open to traffic for a prolonged period, it is more challenging for fog to form. Furthermore, following the plan’s earlier conclusions to enhance the cold light source can significantly reduce any interference caused by fog. This will ultimately improve the accuracy of target marker recognition and effectively protect against any issues of using a VDM for the tunnel project.

## 4. Displacement and Strain Simulation of Railway Tunnel Cross-Section

To effectively evaluate the mechanical properties of the railway tunnel lining structure and select appropriate monitoring points, finite element software Midas GTS NX 2019 was used to conduct two-dimensional modeling of the railway tunnel cross-section. To improve the calculation accuracy and efficiency, the calculation boundary was set to 3 times the equivalent diameter of the tunnel while the range of the model was set at 60.00 m horizontally and 50.00 m vertically. Practical experience shows that when the computational boundary is set to 3 times the equivalent diameter of the tunnel in finite element analysis software, the boundary effect can be effectively controlled [31]. The data used in the software are shown in Table 1. The operating stress of the surrounding rock above the tunnel was calculated. The boundary constraints of the model were horizontal constraints on the left and right sides and vertical constraints on the lower side. Considering the “arching effect” of the surrounding rock, the thickness of the upper surrounding rock of the tunnel model can be determined by the following formula:(4)$$h=\frac{\sigma -c}{k\gamma \tan\Phi }(1-e^{-\tan\Phi })$$
where *h* is the thickness of the surrounding rock, *c* is the cohesion of the surrounding rock, *γ* is the gravity of the surrounding rock, Φ is the angle of internal friction of the surrounding rock, *k* is the coefficient of elastic resistance of the surrounding rock, and *σ* is the stresses applied to the surrounding rock.

The model of the double-lining tunnel structure (see Figure 9a) comprises an inner and outer lining, both made of C25 and C30 concrete, respectively. Before cement filling, the inner and outer lining are connected by a spring member. The lower sides of the tunnel are subjected to transverse loads that are more intense and concentrated than the vertical loads on top. This causes a greater concentration and distribution of force units on the left and right sides of the lower section. Conversely, the vertical loads on top are mostly evenly distributed without significant concentration. An equal loading force is applied to the model, and the corresponding stress–strain and displacement information of the tunnel lining model is displayed in Figure 9b,c following loading.

As illustrated in Figure 9b, the tunnel simulation model indicates that the shear stress on the initial lining of the tunnel is evenly distributed in the vault, accounting for roughly 20%, following the collective loading force. The stress–strain is most noticeable in the second lining of the tunnel. The vault’s distribution increases gradually from the top to the bottom, from the center line. The position of the arch waist and bottom is affected by the bias pressure and experiences the highest stress in the tunnel. The single shear stress on the arch’s bottom can reach up to 200 KN. In the monitoring of tunnels, it is necessary to deploy strain sensors with uniform distribution in the circular cross-section of the tunnel. The strain sensors positioned at the bottom of the arch have a broader range of installation.

As illustrated in Figure 9c, the tunnel’s maximum displacement in the vertical Y-direction is approximately 5.48 mm, representing only 3.1% of the total displacement, mainly concentrated at the tunnel crown. The vertical displacement ranges from 2.7 mm to 3.1 mm, constituting 62% of the total displacement. A total of 5% of the displacement occurred due to the overall loading force and the presence of tunnel bias and high geopathic stresses. The lateral direction (X-direction) of the tunnel experiences displacement that mainly concentrates in the bottom left and right sides of the tunnel. After complete loading, the lateral or X-direction displacements were primarily concentrated at the tunnel bottom’s left and right sides. This phenomenon is due to the high ground stress presence and bias pressure. The lateral movement stayed at 1.47 mm, which was accountable for more than 40% of the total displacement.

## 5. Case Study and Analysis

As is shown in Figure 10A, the tunnel lining structure health monitoring system is mainly composed of a VDM to monitor the crown displacement at the tunnel cross-section and seven pairs of FBG strain sensors which are used to measure the strain variation of the tunnel lining. There are seven measurement points; each point has a pair of FBGs placed perpendicular to monitor both the radial and axial strain of tunnel lining (shown in Figure 10A). FBG strain sensors measuring points 1, 2, 3 and 5, 6, 7 are placed symmetrically along the center line of the tunnel cross-section with heights of 2 m, 4 m, and 6 m from the ground, respectively. The FBG strain sensors can monitor the strain change of the tunnel cross-section and are considered as a supplementary solutions to evaluate the displacement information obtained from VDM. The VDM has a focal length of ~20 m and is able to cover a monitor area of 8 m by 8 m, which includes most of the tunnel cross-section. Due to the limited installation time and the safety requirement in the high-speed railway tunnel, the tunnel linings only below the crown areas are selected. The FBG strain sensors are installed around the tunnel lining cross-section at K468 + 20 m, which is approximately 2.7 km from the tunnel entrance. The data from FBG strain sensors and VDM are transmitted to the FBG interrogator and industrial computer (as shown in Figure 10B) in the tunnel cavern via optical fiber and electrical cables, respectively. An industrial computer is used to process the acquisition data. Once the monitored data exceed the preset threshold, an alarm will be given so as to ensure the tunnel lining of the monitored cross-section is within a safe state.

The reflection rate of the FBG sensor used in this paper is one in ten thousand. With a center wavelength of 1549 nm, falling within the near-infrared spectrum, it is suitable for use in optical communication and fiber optic sensing fields. The applicable temperature range is from −40 degrees Celsius to 80 degrees Celsius, with a wide operating temperature range to meet monitoring needs in harsh environmental conditions.

The sampling frequency of the FBG demodulator used in this paper is 100 Hz, while it has a maximum sensing channels of sixteen. The measurement accuracy of the FBG strain sensor is ±1 µε, which is sufficient for tunnel crown deformation monitoring applications.

### 5.1. Analysis of Fiber Optic Grating Strain Sensor Measurement Point Data

#### 5.1.1. Measured Point Strain Data for No Train Traffic

At randomly selected times when the tunnel is free of train traffic, data from strain sensors at various points on the downstream side were processed and analyzed, as illustrated in Figure 11. In the event of no train traffic, the tunnel ideally reaches a state of silence. During this period, the data should consist of zero-value points in a completely ideal setting, and no fluctuations can be analyzed. It can be observed that, although the strain value of each measurement point is not zero at certain moments, the overall trend of its fluctuations is in the vicinity of zero value with a maximum fluctuation of no more than 0.8 microstrain. The nonzero values could be attributed to environmental disturbances or electronic systems within the tunnel, yet they remain within reasonable parameters. The results proved that the FBG interrogation system is capable of high accuracy strain monitoring of the tunnel lining structure.

#### 5.1.2. Short-Term Tunnel Lining Strain Measurement

Strain variations of the tunnel lining at measurement points 1, 2, 3 for three hours from 12:00 to 15:00 on a particular day were randomly selected, and it was found that trains were passing at 12:51, 13:32, 14:09, and 14:23. Due to the wide range of special trains designed and operated in this railway tunnel, there are several types of trains, such as high-speed trains, ordinary express trains, and freight trains. The real-time data graph of the six FBG strain sensors on the downlink side is shown in Figure 12.

Combined with the data in Figure 12, it can be seen that in the absence of an instantaneous extreme load (earthquake, mudslide, etc.), when there is no train passing, the tunnel structure changes little, the FBG strain sensor value remains at zero fluctuation, and the strain fluctuation of each measurement point is basically the same, and there is no obvious difference. When a train passes through, the passing train will bring additional dynamic load to the tunnel structure due to factors such as load and speed, and the tunnel structure will change accordingly. The specific performance is that the stresses and strains are gradually generated and transmitted to the fiber grating strain sensor. Analysis of the above diagram shows that the maximum single change in stress on the inner wall of the tunnel caused by the passage of a single train is maintained within five microstrains. Further investigation shows that when the train passes, the strain data of No. 1, No. 2, and No. 3 measurement points show a decreasing trend, indicating that the lower the height from the ground, the greater the stress and strain of the measurement points. At the same time, the circumferential strain of No. 1, No. 2, and No. 3 measurement points is greater than the longitudinal strain of the same measurement point, indicating that the circumferential shear stress caused by the passage of the train is greater than the longitudinal shear stress.

#### 5.1.3. 24 h Tunnel Lining Strain Measurement

According to the onsite engineering environment, the continuous 24 h data analysis of the third measuring point at 6 m on the downlink side was randomly selected, and the fitting data are shown in Figure 13. The fluctuation trend of the circumferential sensor and the longitudinal sensor at the third measuring point at 6 m down that day are basically the same. It can be clearly seen that the data change of the circumferential sensor is greater than that of the longitudinal sensor, that is, the circumferential shear stress in the tunnel is greater than that of the longitudinal sensor. The longitudinal stress conforms to the trend of data volatility and the actual stress trend, and meets the theoretical requirements.

#### 5.1.4. 7*24 h Strain Monitoring at Single Measurement Point

According to the field environment, the data of the measurement points at FBG-3 for 7 consecutive days were randomly selected, while the strain variation is shown in Figure 14. It can be clearly observed that the data fluctuation of 7 consecutive days is obvious, and the data trends of longitudinal and circumferential sensors are basically the same. The fluctuation time range of the two is basically the same, without lag phenomenon, and the fluctuation intensity is basically the same and within the theoretical range (±10 microstrains), without obvious deviation. After 7 consecutive days of data acquisition and analysis, it is found that the sensor strain value basically returns to the vicinity of zero value, which indicates that there is a dynamic change in the tunnel structure, but its change is maintained in a weak controllable range, which meets the theoretical needs.

#### 5.1.5. Long-Term Tunnel Lining Strain Measurement

After the stable operation of the system, combined with the local hydrology and geology, the fiber grating strain measurement data for three months from March to June (once a day at a fixed time) were selected to analyze the long-term stability of the fiber grating strain sensor. The specific data trends are shown in Figure 15. It can be seen that during the three months of normal operation of the tunnel, the strain change trend of each measuring point of the tunnel is basically the same, that is, there is no difference in the structural state of the tunnel; further analysis shows that due to the influence of the gravity load of the tunnel structure, the figure data show that the stress and strain generated by the measuring points with lower distance from the ground are larger, and the circumferential strain is greater than the longitudinal strain, which is also consistent with the strain trend of the measuring points during short-term passing. It is clear from the figure that there is no obvious mutation in the overall strain value, and the tunnel structure is safe and reliable. It also shows that the optical fiber monitoring system and the sensor can work stably in the wet and dark environment of the tunnel.

### 5.2. Optical VDM Measurement Analysis

We set the video displacement meter 20 m away from the measuring point and 1 m above the ground, and set up two fill lights, with the side of fill light 1 as measuring point A and the side of fill light 2 as measuring point B, respectively, corresponding to measuring points 3 and 2 of the FBG strain sensor monitoring the settlement displacement of the tunnel, as depicted in Figure 16.

#### 5.2.1. Short-Term Tunnel Lining Displacement Monitoring

The real-time data of the downside VDM for 30 min from 10:00 to 10:30 on a particular day were randomly selected. The analysis showed that trains started passing at 10:10, 10:19, and 10:26 passed through the cross-section of the measurement point, as shown in Figure 17. The analysis shows that when there is no train passing through the monitoring surface, the data of the tunnel measuring point are basically maintained near the zero point, and the fluctuation is small; when a train passes through the monitoring section, the displacement of the measuring point changes obviously, and the displacement decreases rapidly after the train passes through, which is in line with the theoretical state of the tunnel before and after the dynamic load is applied. At the same time, further analysis shows that there are also differences in the displacement disturbance of the tunnel caused by different vehicle speeds or models. There is a certain fluctuation in the displacement data (see Table 2 of the fluctuation data), and it finally returns to the original state. The displacement waveform data correspond to the theoretical short-term arrival. When the train passes, the tunnel structure returns to the original state before the fluctuation.

#### 5.2.2. 24 h Displacement Data at a Single Measurement Point

According to the engineering environment on site, the data analysis of continuous 24 h of the randomly selected B measurement point is shown in Figure 18. We can clearly observe and conclude that the traffic is heavy in the tunnel that day while different train weights and speeds caused different displacement of the measuring points. The fluctuations of the displacement conformed to the repeatability rule (when a single vehicle passed, the lateral displacement of measuring point B basically changed by 16.00 mm, and the vertical displacement of measuring point B basically changed by 10.00 mm), and there are obvious differences in the monitoring points when there is or is not a car passing through the tunnel, which satisfies the theoretical analysis.

#### 5.2.3. Single Measuring Point Continuous 7*24 h Displacement Data

Figure 19 shows the data of randomly selected monitoring point B for 7*24 h. We can clearly observe and conclude that the number of traffic in the tunnel is high within 7*24 consecutive hours, and the displacement values caused by different vehicles and speeds (including bullet trains, general express vehicles, and trucks) in the same direction of the same measuring point are also different, and there is fluctuation of data. The range value is more in line with the repeatability law (the longitudinal displacement of measuring point B basically changes within 10 mm each time a vehicle passes), and there are obvious differences in the monitoring points when there is an incoming vehicle on the track or not, which corresponds to the theoretical analysis.

#### 5.2.4. Long-Term Displacement Monitoring

After the system is running stably, combined with the local hydrology, geology, and other conditions, we selected the three-month optical VDM measurement data from March to June (taking the data once a day at a fixed time), and carried out an analysis of the long-term stability of the optical VDM; the specific data trend is shown in Figure 20.

Combined with the three-month long-term stability data of the optical VDM in Figure 20, it can be seen that during the operating period, the dynamic load of each monitoring point in the tunnel will cause the corresponding displacement of the measuring point to change when the tunnel is opened to traffic, and the trend of the displacement of the measuring point remains constant. This means that the displacement of the measurement points fluctuate around zero and no cumulative displacement change occurs, i.e., the rock stress and load stress do not cause any actual displacement changes in the tunnel structure, which meets the theoretical analysis.

Furthermore, it can be seen that from the combination of actual monitoring data and the simulation model that the type and frequency of passing trains will cause changes in the instantaneous stress, strain, and displacement of the tunnel. Among these, the opening of traffic will cause a short-term slight displacement on the inner surface of the tunnel, but the tunnel will basically recover after the train has passed. Under the same circumstances, as the train passes through the tunnel, it will cause stress and strain, and the release of stress and strain is relatively slow, with a certain hysteresis, but when the stress is released, the strain also returns to the initial state. The expected results obtained from the two optical sensor monitoring methods are basically consistent, which preliminarily shows that the passenger train will not affect the structural state of the tunnel’s stable operation. The stress and strain of the tunnel under high ground load are mainly influenced by the gravity load and the structural distortion. Considering that the current stress and strain of each measurement point are within a reasonable range and the current data samples are limited, the effect of the tunnel monitoring data on the bias stress will be continuously monitored in the future. More detailed analysis will be undertaken.

## 6. Conclusions

In this paper, the FBG strain sensors were used to monitor the microstrain level of the railway tunnel lining. The strain of the tunnel section in the passing train is mainly reflected in the position with the lowest height from the ground, which is the most strained and basically maintained within 5 microstrains. The hoop strain at the measurement point is 1~2 microstrains greater than the longitudinal strain. The accuracy of the optical VDM can reach submillimeter-level displacement monitoring. The displacement and settlement of the tunnel section in the passing train is mainly reflected in the vertical direction. The position with the highest height from the ground is most affected by the displacement, and the instantaneous displacement of the measuring point has the largest change during multitrain operation within 17.90 mm, and the vertical displacement changes by 10.00 mm. The data shows that under long-term operating conditions, the displacements of the measurement points in the tunnel do not increase in one direction. At the same time, the high-precision and multipoint monitoring using an integrated monitoring system with FBG and VDM can provide real and effective data for the structural evaluation of tunnels, and provide reference and guidance for the health monitoring and maintenance of related railway tunnels and other structures. At the same time, the technology can provide real and effective data for the structural evaluation of tunnels, and can provide guidance for the health monitoring and maintenance of related railway tunnels and other structures.

## Figures and Tables

**Figure 1 sensors-24-06201-f001:**
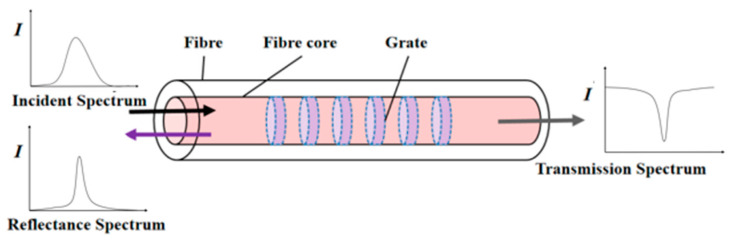
Principle of FBG sensors.

**Figure 2 sensors-24-06201-f002:**
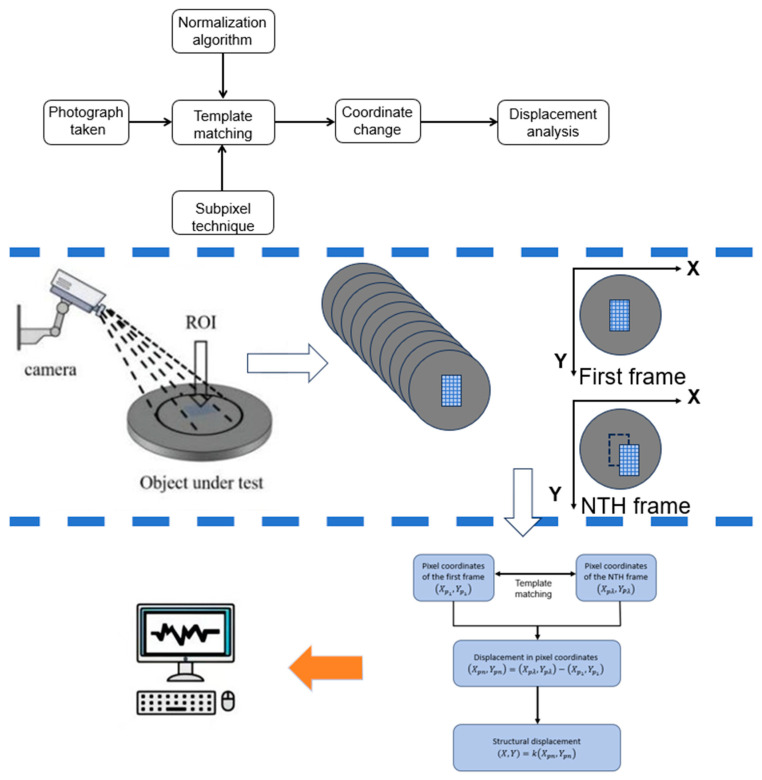
Normalized correlation algorithm flow diagram.

**Figure 3 sensors-24-06201-f003:**
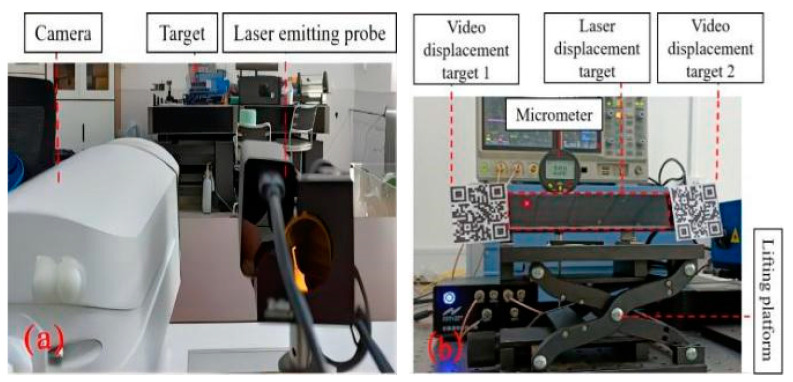
Displacement meter comparison experiment diagram.

**Figure 4 sensors-24-06201-f004:**
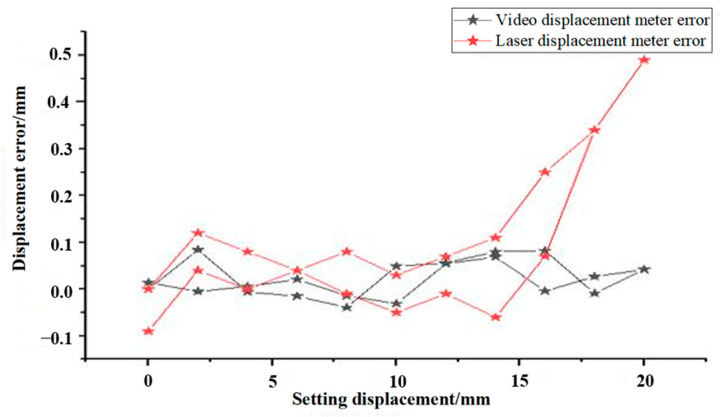
Comparison of the accuracy of the two types of displacement meters.

**Figure 5 sensors-24-06201-f005:**
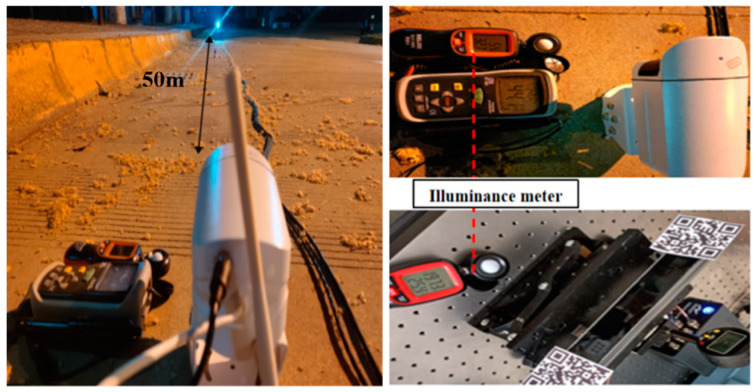
VDM light level experiment diagram.

**Figure 6 sensors-24-06201-f006:**
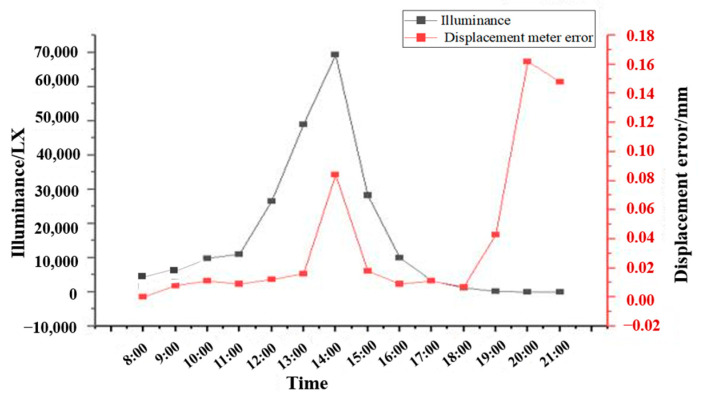
VDM illuminance data diagram.

**Figure 7 sensors-24-06201-f007:**
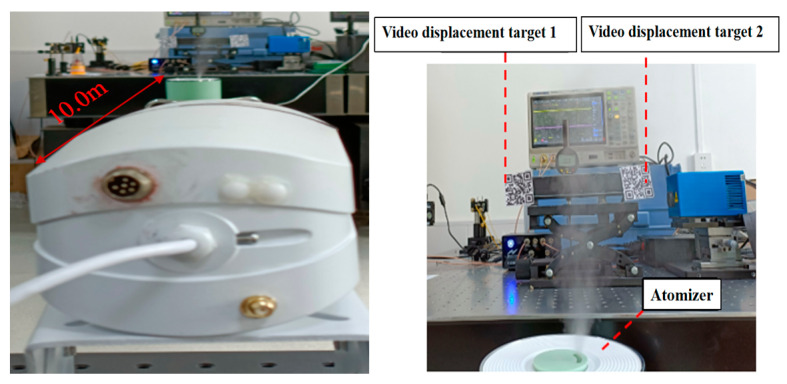
Diagram of VDM for fog experiment.

**Figure 8 sensors-24-06201-f008:**
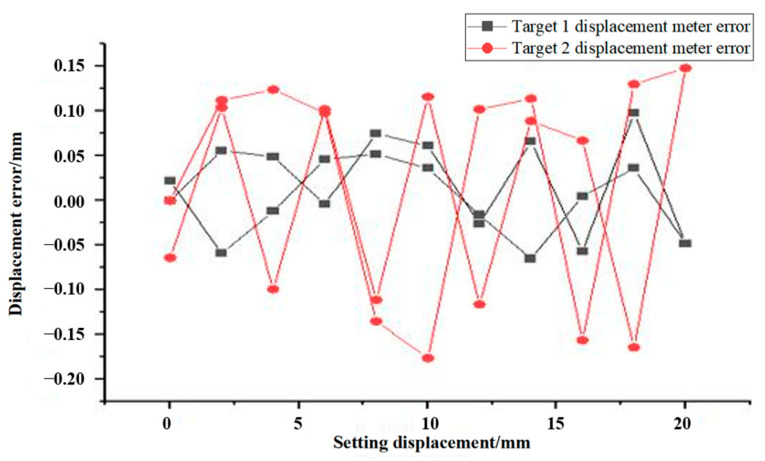
VDM fog data error diagram.

**Figure 9 sensors-24-06201-f009:**
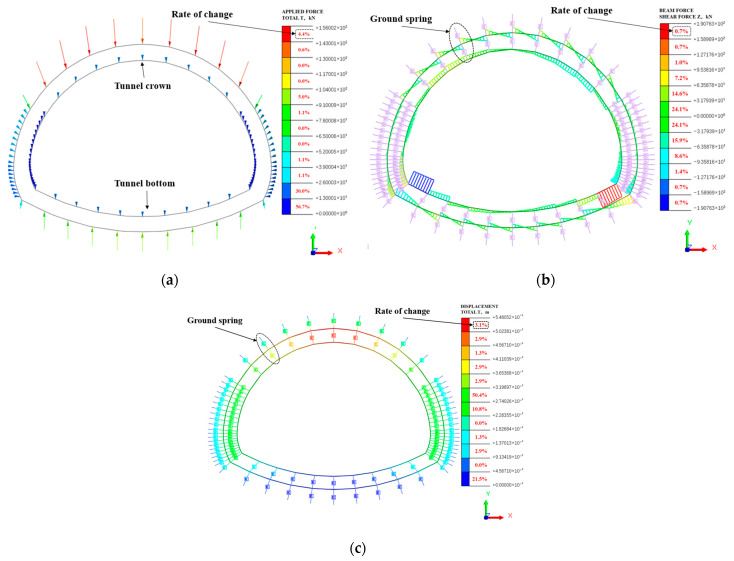
Simulation of stress distribution of double lining tunnel structure model, (**a**) double lining tunnel structure model, (**b**) equivalent load stress–strain diagram, (**c**) equivalent load displacement diagram.

**Figure 10 sensors-24-06201-f010:**
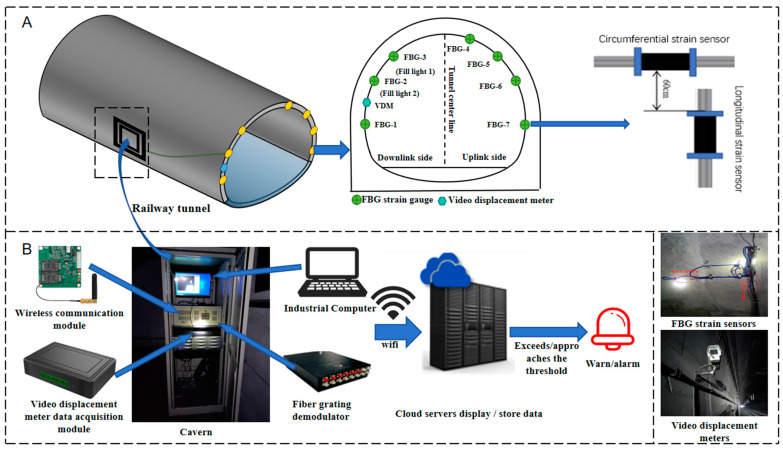
Topology diagram of tunnel structure safety monitoring system.

**Figure 11 sensors-24-06201-f011:**
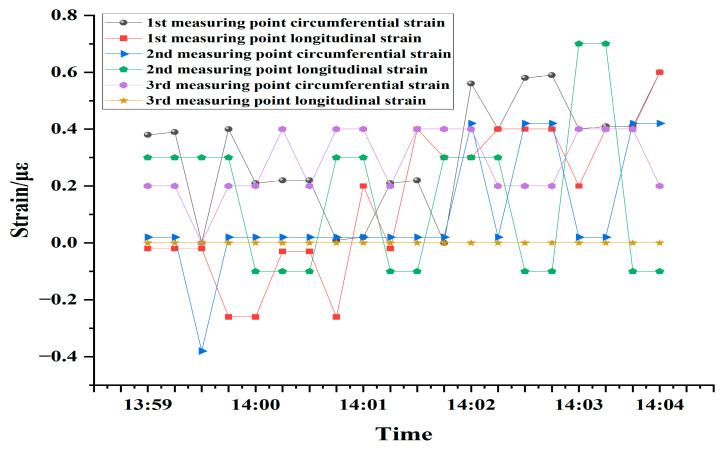
FBG strain sensor output versus time without passing trains.

**Figure 12 sensors-24-06201-f012:**
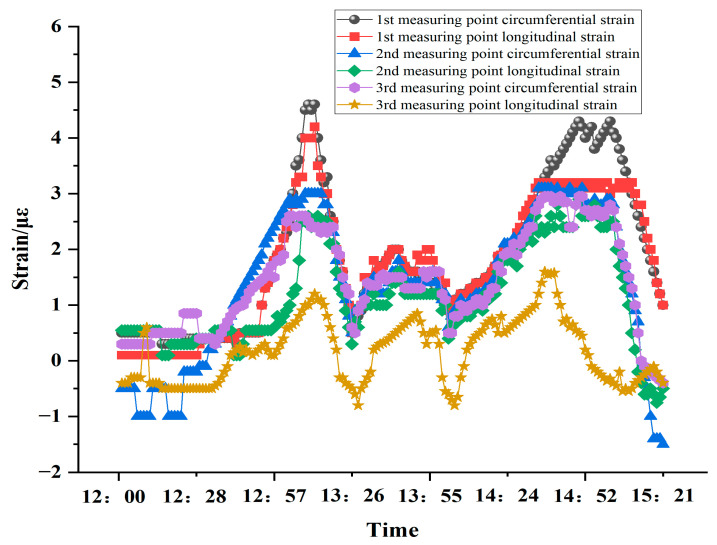
FBG strain sensor traffic data diagram.

**Figure 13 sensors-24-06201-f013:**
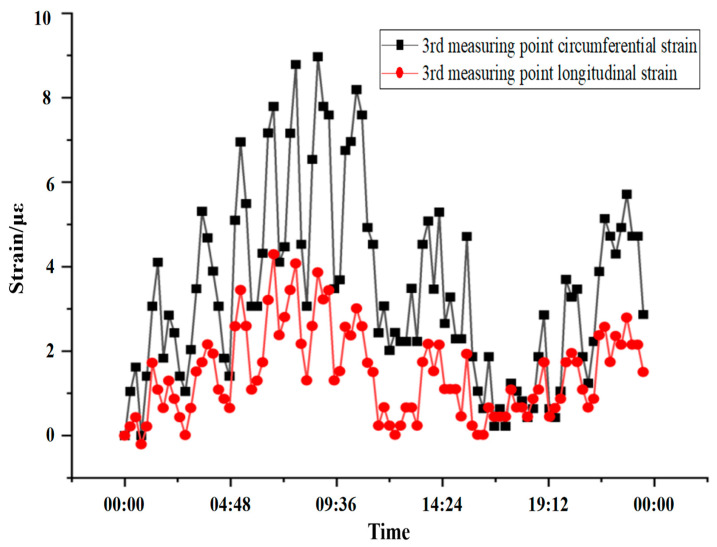
Continuous 24 h data diagram of the measuring point at FBG-3.

**Figure 14 sensors-24-06201-f014:**
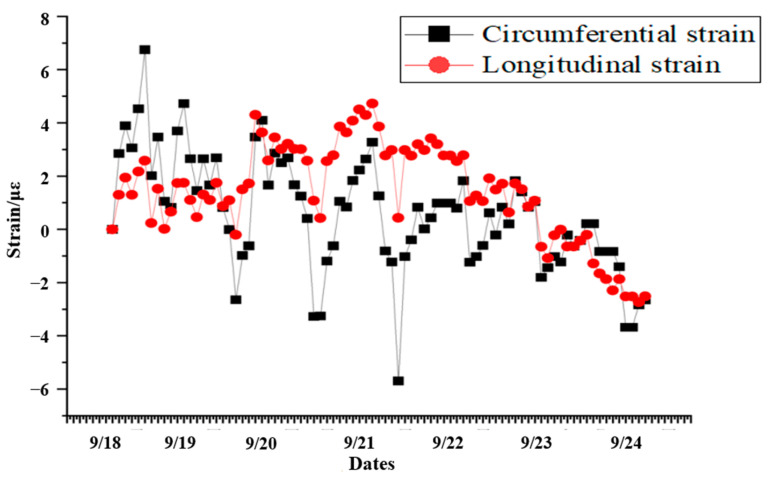
The data diagram of the measuring point at FBG-3 for 7 consecutive days.

**Figure 15 sensors-24-06201-f015:**
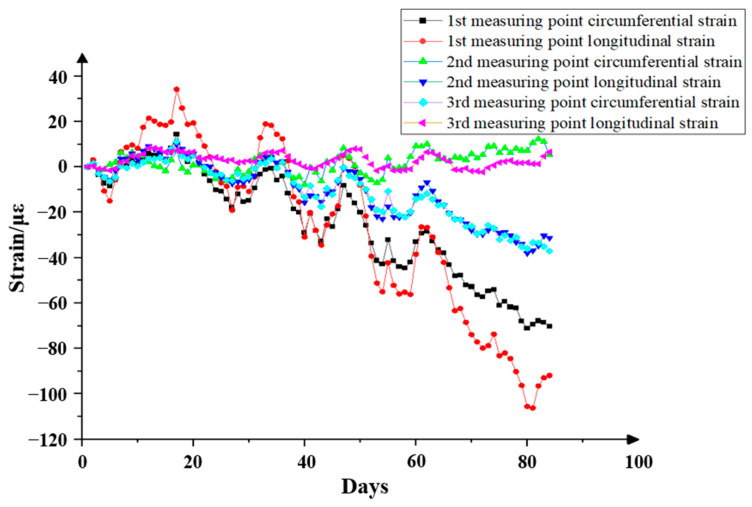
Long-term data diagram of strain at the measuring point on the downlink side.

**Figure 16 sensors-24-06201-f016:**
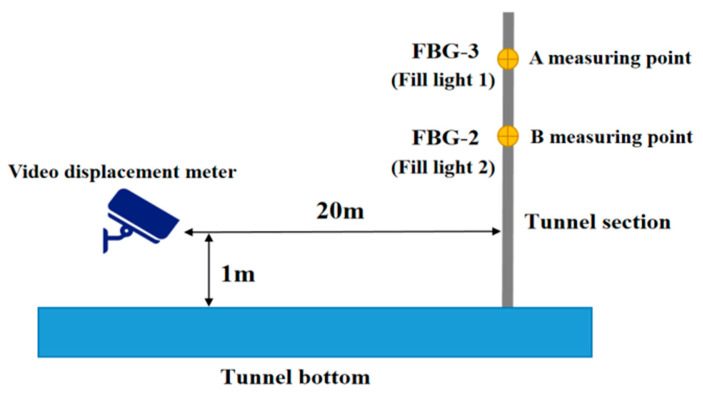
VDM installation schematic.

**Figure 17 sensors-24-06201-f017:**
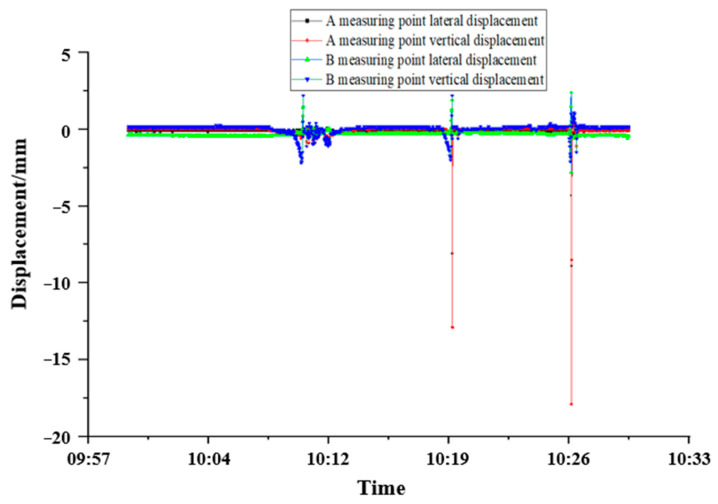
Short-term data trend diagram of optical VDM.

**Figure 18 sensors-24-06201-f018:**
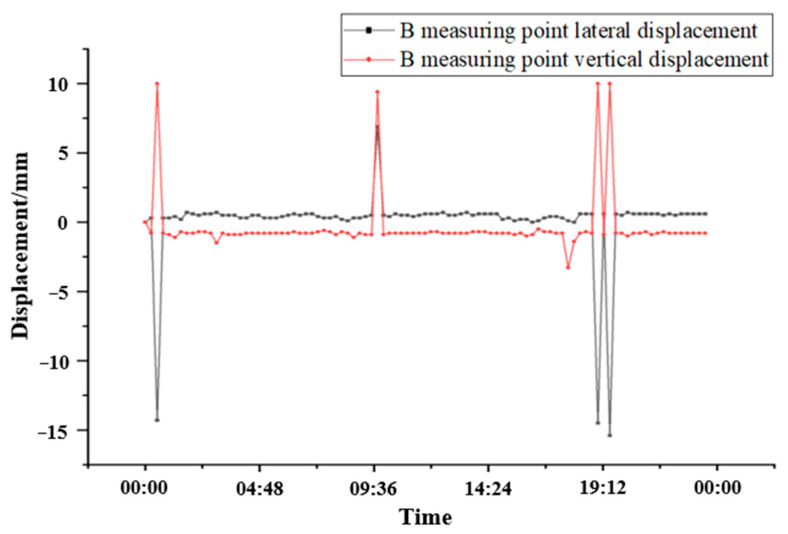
Continuous 24 h displacement data diagram of measurement point B.

**Figure 19 sensors-24-06201-f019:**
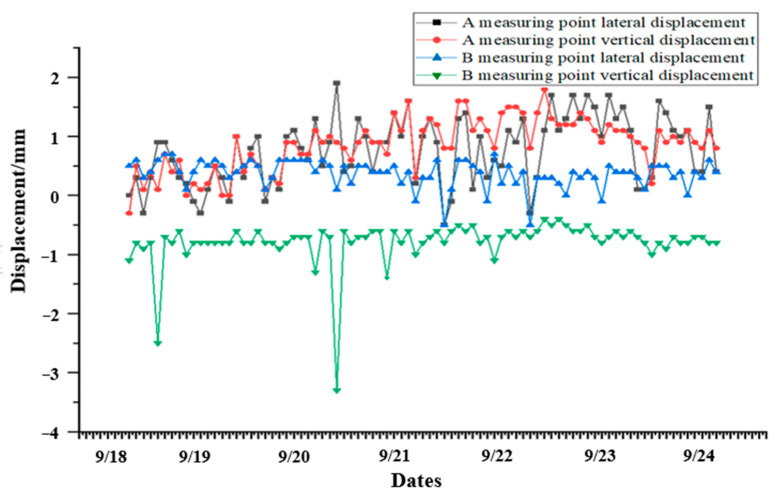
Continuous 7*24 h displacement data trend diagram of measurement point B.

**Figure 20 sensors-24-06201-f020:**
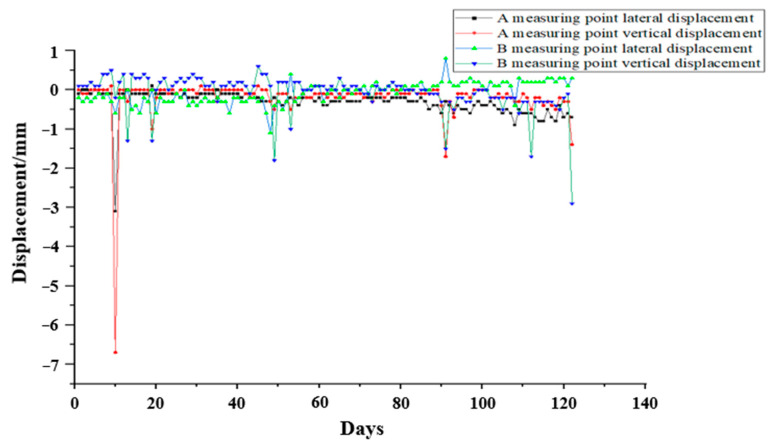
Long-term stability data trend diagram of VDM.

**Table 1 sensors-24-06201-t001:** The model draws up the mechanical parameters of the surrounding rock.

Symbol	Quantity	Value
*E* _1_	Young’s modulus of rock and soil	1.1–5 GPa
*E* _2_	Young’s modulus of lining	35.5 GPa
*μ* _1_	Poisson’s ratio of rock and soil	0.2–0.3
*μ* _2_	Poisson’s ratio of lining	0.2
*c*	Cohesion	0.28–0.35 MPa
*Φ*	The angle of internal friction	50–60°
*γ* _1_	Gravity of rock and soil	15–18 kN/m^3^
*γ* _2_	Gravity of lining	25 kN/m^3^

**Table 2 sensors-24-06201-t002:** Statistical table of displacement fluctuations before and after the train passes.

Measuring Point Information	A Measuring Point (mm)		B Measuring Point (mm)	
	Lateral Displacement	Vertical Displacement	Lateral Displacement	Vertical Displacement
Before opening to traffic	−0.10	0.00	−0.40	0.10
First train a	0.00	−0.50	1.50	2.20
Second train b	−8.10	−12.90	−0.60	−0.60
Third train c	−4.30	−17.90	−2.80	−0.50
After opening to traffic	−0.10	0.00	−0.50	0.20

## Data Availability

The data are confidential.

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
