# Peer review of "Simultaneous Measurement of Strain and Displacement for Railway Tunnel Lining Safety Monitoring"

_sensors, 2024, doi:10.3390/s24196201_

Round 1
Reviewer 1 Report
Comments and Suggestions for Authors
The authors presented a sensor system based on fiber Bragg gratings and video displacement measurement for monitoring railway tunnel lining. The authors promised in the abstract a comprehensive experimental study of FBGs for strain measurement. But then they showed what recorded particular FBG sensors during the passage of the trains and without traffic. I cant see any comparison with an electrical sensor. The abstract should be reformulated to define better the text of the manuscript.
In the introduction should be not only named particular solutions but also their advantages and drawbacks and parameters of for example resolution, range.
As far as I can see, only a small part of the tunnel lining can be monitored with this method. Why should I use this system? In addition,, the authors didnt show that the system is working. They showed that there is no change if there is no any kind of change, except traffic.
Comments and small corrections
row 60 typo analyise
row 80 Capital letter the measurement
using abbreviation without introducing VDM
row 173 not above but bellow
row 168 it should write the date of making experiment , because in spring is less light than in summer.
row 241 what do you mean by 3-5 times?
row 292 by 8 by 8 m you mean 64 square meters?
After the equation there should be a dot or comma, it depends.
In Fig 4 there is different character depending on if distance isincreasing or decreasing. Why? Also for values less than 14 mm the error is preaty same, but for last thre values it is increasing. Why?
Comments on the Quality of English LanguageEnglish is fine but it is necessary to improve text.
Reviewer 2 Report
Comments and Suggestions for Authors
This article employs FBG interrogation and VDM systems to monitor the strain variation and deformation of the cross-section of a railway tunnel. The article is well organized. I have a few questions and comments that may require further clarification from the authors:
1. In the title, “Simultaneous” is the correct adjective form to describe the measurement process.
2. The full names of FBG and VDM should be provided in both the abstract and introduction when the abbreviations are first mentioned.
3. In Figure 1, the color distributions of the incident and reflected spectra appear incorrect to me.
4. The specific brand, model, and key parameters of the commercial FBG interrogator used should be provided.
5. In line 307, the sensitivity in an FBG sensor is more related to how small changes in the physical environment (such as strain, temperature, or pressure) can shift the wavelength of the reflected light, rather than the amount of light that is reflected. The 0.01% reflection rate does not directly imply high sensitivity that depends on the design of the FBG, the properties of the fiber, and the interrogation technique used.
6. In Figure 11, how do you distinguish between longitudinal and axial strain at a specific measurement point?
7. Various optical techniques such as OFDR exist for distributed strain measurements with higher spatial resolution. What led to the choice of FBG sensors in this study? Incorporating insights from additional sources, such as those suggested (10.1364/OL.504763, 10.1109/JSEN.2022.3197730, etc.), would offer readers a more comprehensive understanding of the subject matter.
Please consider and use these comments as my support to improve your paper.
Comments on the Quality of English LanguageThe presentation is generally okay but will benefit from a thorough review to correct grammatical errors.
Round 2
Reviewer 1 Report
Comments and Suggestions for Authors
The authors answered my questions satisfactorily.
Reviewer 2 Report
Comments and Suggestions for Authors
My questions and concerns have been addressed. While there are some formatting and language issues, these can be refined during proofreading.
Comments on the Quality of English LanguageMinor polishing is suggested with proofreading.